# Enhancing Magnetic Hyperthermia Nanoparticle Heating Efficiency with Non-Sinusoidal Alternating Magnetic Field Waveforms

**DOI:** 10.3390/nano11123240

**Published:** 2021-11-29

**Authors:** Michael Zeinoun, Javier Domingo-Diez, Miguel Rodriguez-Garcia, Oscar Garcia, Miroslav Vasic, Milagros Ramos, José Javier Serrano Olmedo

**Affiliations:** 1Center for Biomedical Technology (CTB), Universidad Politécnica de Madrid (UPM), Campus Montegancedo, 28233 Madrid, Spain; javier.domingo@ctb.upm.es (J.D.-D.); miguelramon.rodgarcia@alumnos.upm.es (M.R.-G.); milagros.ramos@ctb.upm.es (M.R.); 2CIBER de Bioingeniería, Biomateriales y Nanomedicina (CIBER-BBN), 28029 Madrid, Spain; 3Centro de Electrónica Industrial, Universidad Politécnica de Madrid (UPM), 28006 Madrid, Spain; o.garcia@upm.es (O.G.); miroslav.vasic@upm.es (M.V.)

**Keywords:** hyperthermia, magnetic nanoparticles, superparamagnetic, iron oxide, nanomedicine, alternating magnetic field, non-harmonic waveforms

## Abstract

For decades now, conventional sinusoidal signals have been exclusively used in magnetic hyperthermia as the only alternating magnetic field waveform to excite magnetic nanoparticles. However, there are no theoretical nor experimental reasons that prevent the use of different waveforms. The only justifiable motive behind using the sinusoidal signal is its availability and the facility to produce it. Following the development of a configurable alternating magnetic field generator, we aim to study the effect of various waveforms on the heat production effectiveness of magnetic nanoparticles, seeking to prove that signals with more significant slope values, such as the trapezoidal and almost-square signals, allow the nanoparticles to reach higher efficiency in heat generation. Furthermore, we seek to point out that the nanoparticle power dissipation is dependent on the waveform’s slope and not only the frequency, magnetic field intensity and the nanoparticle size. The experimental results showed a remarkably higher heat production performance of the nanoparticles when exposed to trapezoidal and almost-square signals than conventional sinusoidal signals. We conclude that the nanoparticles respond better to the trapezoidal and almost-square signals. On the other hand, the experimental results were used to calculate the normalized power dissipation value and prove its dependency on the slope. However, adjustments are necessary to the coil before proceeding with in vitro and in vivo studies to handle the magnetic fields required.

## 1. Introduction

The cancer burden is increasing exponentially. It was estimated to have risen to 18.1 million new cases and 9.6 million deaths worldwide in 2018 [1]. The World Health Organization (WHO) estimated that 1/5 men and 1/6 women worldwide develop cancer and 1/8 men and 1/11 women die from the disease [2]. The most common treatments of cancer are surgery, radiotherapy and chemotherapy. However, the lack of efficiency of these treatments in many cases, in addition to their side effects that have not yet been totally neutralized, indicates that the search for new cancer treatments and therapies is vital. One of the most interesting approaches is hyperthermia, which is a non-invasive technique to selectively kill target cancer cells by raising the microenvironment tumor temperature to 43 °C to 45 °C, resulting in cell apoptosis due to the irreversible cell damage caused by their low heat resistance, where on the other hand, normal cell damage is minimum due to their high resistance to heat [3,4].

There are various hyperthermia techniques, with only one of them currently being clinically used. It consists of using antennas at microwave frequencies combined with radiotherapy [5] and chemotherapy [6]. Hyperthemia showed various therapeutic benefits, such as some direct cytotoxicity due to heating, mild anti-tumor response [7], and tumor sensitization [5]. However, this work focuses on the nanomedical cancer treatment hyperthermia, or magnetic hyperthermia (MHT), where the temperature increase is mediated by magnetic nanoparticles (MNPs) [8]. It consists of placing superparamagnetic NPs under the influence of an alternating magnetic field (AMF), resulting in heat generation due to Néel and Brownian relaxation losses [3]. The AMF, however, is generated through a coil connected to an AMF generator. It is essential to point out that AMFs applied at frequencies and intensities above 1 MHz and 1 T could be harmful to tissues and biological fluids [9], which is not the case of the experiments in this study.

Nanoparticles (NPs) with different properties, such as gold, silver, or iron oxide, are commonly used by the scientific community to develop new applications, such as therapies or medical diagnostics [10]. MNPs are well known and have many applications in the biomedical field: drug delivery [11], tissue engineering [12], magnetic resonance imaging (MRI) contrast agent [8,11] or heat mediators in optical hyperthermia [10]. Iron oxide is the most commonly used material due to its high compatibility and biodistribution in the body [13]. Furthermore, MNPs have been approved by the food and drug administration (FDA) for some applications, such as MRI contrast agents and heat mediators for drug release or cancer treatment using AMF [13], as we previously mentioned. Furthermore, the iron oxide NPs are the most promising heating agent candidates [14], which is why they were used in the MHT clinical trials of Magforce AG to test their NanoTherm^®^ therapy system [15]. There are many different sizes and shapes of these NPs. The particle size affects its heat production because, theoretically, the size at which the magnetite particles change from Néel to Brown is around 12 nm [16]. Thus, if the MNP is smaller than 12 nm, Néel Relaxation dominates its heating process; hence, it is dominated by frequency. On the other hand, if the MNP is larger than 12 nm; it is dominated by Brownian relaxation; hence, the heating generation depends on the AMF strength [16]. The shape of the MNP also affects its performance, and there are many different iron oxide NPs, such as nanoflowers, spherical and hollow [16,17]. The worldwide scientific network is in a constant influx of information and basic research to obtain improvements and to consolidate developments cooperating in the advancement of MHT to obtain outcomes in clinical translation [14].

In Sánchez et al. [18], the cubic-shaped NPs were discussed, followed by their optimization for MHT application and their potential industrial implementation. They showed good heating performance, making them a promising type of particles for MHT. Duta et al. [19] showed the development of malic acid and α-hydroxy acid grafted magnetic nanocarriers of spherical size, good magnetic responsiveness and excellent heating efficiency. As shown in Gustafson et al. [20], it is vital for drug delivery agents to possess good biocompatibility and avoid the rapid clearance from the body by the reticuloendothelial system (RES), in addition to inertness and high loading affinity of drug molecules. The iron oxide superparamagnetic nanoparticles (SPIONs) showed a high specific absorption rate (SAR), a 72% loading efficiency and a slow and sustained release in a pH-dependent manner over a period of time, making them suitable for MHT and controlled drug delivery. In Mille et al. [21], the NP chain formation during MHT through the calorimetric method was studied using time-resolved high-frequency hysterisis loops. The study proved that the formation of the chains is dependent on time and magnetic field amplitude, whereas it is not dependent on the frequency in the range they experimented (between 9 and 78 kHz). The work also concluded that if the measurement time is shorter than the chain formation, the thermal power is underestimated. A fact that is vital for biological applications since the magnetic fields are applied for a longer duration than the heating power measurements. Similarly, in Iglesias et al. [22], MHT through calorimetric methods was conducted in order to enhance the power losses calculation and clarify some irregularity parameters.

In this study, we will be using spherical iron oxide NPs, with a diameter of approximately 10.6 nm at two different concentrations (38.6 and 75 mgFe/mL). It is expected to have a difference in heating, where the heating efficiency decreases by half when going from the latter to the former concentration. Therefore, we will observe differences in heating efficiencies depending on the concentration and other variables explained in the Materials and Methods section.

### 1.1. Hyperthermia Limitations

Nevertheless, MHT faced various limitations, such as a significant efficiency decrease between in vivo and in vitro experiments and aggregation. In vivo MHT was described in Manuchehrabadi et al. [12], demonstrating that the homogeneity of the temperature increase inside the organ was strongly dependent on the distribution of NPs. Furthermore, MHT experiments demonstrate that MNPs have a decreased energy absorption per unit mass, or SAR, once internalized in cells, ranging from 90% to 50% depending on particle size, shape and composition [23,24]. The aggregation effect, on the other hand, was demonstrated in Mejías et al. [11], where the inefficiency of NPs to respond to AMF was observed due to NP massive aggregation, which led to the unpredictable NP orientation in tumor cell lysosomes. On the other hand, other research links this effect to Brownian mobility restriction [24,25].

As a result, greater efficacy in producing heat might alleviate most of the issues mentioned above, either by enhancing NP behavior, which is the most widely pursued area of research, or by enhancing energy delivery from the exciting magnetic field, which is the primary goal of this work.

To the best of the author’s knowledge, there are no experimental studies that use non-conventional AMF waveforms or systems capable of providing these waveforms. Thus, all MHT experiments in the articles are conducted using the conventional sinusoidal waveform.

### 1.2. Hypothesis

Rosensweig’s model calculates the NPs’ heat dissipation following its excitation by an AMF [26]. The power dissipation formula is expressed by:(1)$$P_{dissipated}=\pi \cdot \mu_{0}\cdot \chi_{0}\cdot H_{ac}^{2}\cdot f\cdot \frac{2\cdot \pi \cdot f\cdot \tau }{1+{\left(2\cdot \pi \cdot f\cdot \tau \right)}^{2}}$$
where $\mu_{0}$ (H· m${}^{-1}$) is the vacuum permeability, $\chi_{0}$ is the equilibrium susceptibility, *f* (s${}^{-1}$) is the frequency, $H_{ac}$ (A· m${}^{-1}$) is the amplitude of the magnetic field and τ (s) is the relaxation time. However, the model was made for sinusoidal AMF waveforms, and no further models were developed using other waveforms.

Looking at Equation (Equation 1), Rosensweig connected the magnetic field and frequency to the temperature rise after a certain amount of time (ΔT) in NPs [26], meaning that the heat dissipation of the NPs is dependent on *B* and *f*. It also depends on $\chi_{0}$; however, since we are using the same NPs throughout the experiments of this study, the $\chi_{0}$ will not vary.

In initial studies conducted in our laboratory [27], we observed that the heat production does not follow the Rosensweig rule for low magnetic intensities when using non-sinusoidal waveform and non-coupled slopes and peak intensities [26]. Then, in [28,29], which was a continuation of the initial work, the results showed that, contrary to the Rosensweig rule, the frequency is not enough to define the heat production when all the other experimental conditions are kept equal. Furthermore, modifying the heat generation using only square waveforms of shifting duty cycles is conceivable, pointing to transients around the zeros as the primary source of heat dissipation once again. This means that the slope of the AMF signal, signal frequency and even particle size (tiny particles <10 nm) conditions the proportionality between ΔT and *B*.

Therefore, this paper will conduct experiments with a newly developed and enhanced AMF generator, which can reach higher frequencies, generate conventional and various unconventional AMF waveforms and produce higher intensity alternating currents (AC).

### 1.3. Objectives

The objective of the paper is to:Conduct AMF magnetic field strength, frequency and particle concentration experiments to confirm the results obtained previously in Rosales et al. [27], Garcia et al. [28] and Urbano et al. [29], which revealed that the proportionality between ΔT and *B* is conditioned by the AMF signal shape, frequency and particle size.Demonstrate the dependency of ΔT on the AMF waveform slope factor.Establish a thermal power efficiency percentage ratio between conventional and unconventional AMF waveforms.

## 2. Materials and Methods

### 2.1. Materials

The MHT system used in this study consists of an AMF generator connected to a coil, inside of which there is the sample holder. The latter is connected to a water pump so that a continuous flow of water is pumped through. As for the sample’s temperature measurement, we used fiber optic sensors to detect and plot the temperature rise and fall curves (see Figure 1).

#### 2.1.1. AMF Generator

The AMF generator used for the experiments of this article is a full bridge-inverter-based device [28] designed and developed in our laboratory. It is able to generate 4 unconventional signals in addition to a 5th sinusoidal (SN) for comparison (see Figure 2). The 4 unconventional signals are:Triangular (TR);Trapezoidal-Triangular (TT);Trapezoidal (TP);Trapezoidal-Square (TS).

The initial aim was to generate a pure square signal; however, this proved extremely difficult in inverter applications, so an “almost-square signal” was generated, which is TS. The AMF can generate up to 100 $A_{pk-pk}$ output current and operate at frequencies of 100 kHz to 1 MHz at a 100 kHz frequency step.The design is thoroughly described in Zeinoun et al. [30].

The coil used in the experiments is optimized in Garcia et al. [28]. Unfortunately, however, it showed severe overheating problems, making it hard to operate above 30 $A_{pk-pk}$, which is the peak to peak alternating current (AC) generated by the AMF generator, and is equivalent to an AMF intensity of 3.2 mT. Nevertheless, this was enough for the scope of this study since our objective is to determine the difference in heating efficiency between conventional and unconventional AMF waveforms.

#### 2.1.2. Fiber Optic Measurement

The experimental tubes used in the experiments are 177.8 mm × 5 mm tubes, making it hard for conventional temperature measurement tools to reach the sample. Furthermore, since the sample is inside an AC magnetic field, the accuracy of electronic measurement will be affected by the AMF excitation. Hence, the Luxtron 3300 fiber optic temperature measurement tool was used, which has 4 output connections that simultaneously and independently measure and plot the temperature curves on the user’s interface, allowing us to study the temperature variation.

In our experimental protocols, three measurements are taken simultaneously, as shown in Figure 3. Looking at the SPION tube, we notice a Pasteur pipette placed inside the sample tube. It is to keep the fiber optic right in the center of the sample because, based on Wildeboer et al. [31], the thermal conductivity variate between the center, lower and higher parts of the sample. Furthermore, since the fiber optics are eventually cables that might curve, we decided to place a Pasteur pipette to keep it straight and fixed in the middle of the sample.

Therefore, looking at Figure 3, we notice that the sample tube contains two fiber optics. One is going down to the sample (in Red), and the other measures the temperature of the Pasteur pipette (in Green). The reason for the pipette temperature measurement is to detect the temperature surrounding the sample tube for external thermal interference coming from the coil or elsewhere. The third fiber optic measures the room temperature out of a separate cup containing water (in Blue).

#### 2.1.3. Calorimeter

To measure the temperature variation of NPs accurately, the sample must be isolated from external thermal interference, especially in this study, since we faced severe coil heating. Hence, a 3-layer sample holder (referred to from here on out as calorimeter) is designed to isolate the sample. The three layers consisted of a water layer, where 37 °C water is continuously pumped, a vacuum layer and a sample layer [27,28]. We placed a 500 μL de-ionized the water sample inside the calorimeter to test the isolation effect. A SN 25 $A_{pk-pk}$ AC current output equivalent to 2.67 mT AMF intensity was generated at 500 kHz; the results are seen in Figure 4.

The blue, orange and gray curves represent the sample, Pasteur pipette and ambient temperature. The results show that the Pasteur sensor is highly affected by the coil temperature outside the calorimeter, yet the sample’s temperature is fixed at 37 °C. The experiment in Figure 4 was stopped at 800 s because the coil’s temperature reached 150 °C on the thermal camera. This proves that the sample is totally isolated from external thermal stressors; hence, any temperature variation detected inside the sample is generated by the rotating NPs.

#### 2.1.4. Nanoparticles Used

The NPs used are 10 nm diameter superparamagnetic iron oxide nano-spheres without any biofunctionalization, with a hydrodynamic diameter of 47.6 nm and a Polydispersity index (PDI) of 0.23. The isoelectric point is at pH 7.9, whereas the colloid is at pH 3, with a corresponding $\zeta_{potential}$ equal to +21 mV. The transmission electron microscope (TEM) images of the nanoparticles is seen in Figure 5.

They were prepared following the Massart coprecipitation method [32]. A mixture of 445 mL of FeCl${}_{3}\cdot 6$H${}_{2}$O (0.09 mol) and FeCl${}_{2}\cdot 4$H${}_{2}$O (0.054 mol) was slowly added to 75 mL of 3.4% NH${}_{4}$OH at room temperature and under vigorous magnetic stirring. Next, the residue was washed three times with distilled water and separated by magnetic decantation. Finally, particles were subjected to an acid treatment with HNO${}_{3}$ (2 M) for 15 min, then $\mathrm{Fe}(\mathrm{NO}{}_{3}{)}_{3}$ (1 M) was heated up to the boiling temperature under stirring, and finally, once the mixture was cooled down, it was treated again with HNO${}_{3}$ (2 M) [33].

The resulting sample was washed again three times with distilled water, separated by magnetic decantation and redispersed in water under sonication for 15 min, leading to a colloidal suspension with a concentration of 75 mgFe/mL.

No biofunctionalization was used in this article because the objective is the behavior of the NPs under different waveforms and the effect of the slopes on the performance of the thermal power, and biofunctionalization may affect that at this time. However, before starting in vitro experiments, the inert experiments described in the manuscript will be repeated using biofunctionalized NPs to compare inert and biological tissue results.

### 2.2. Methods

#### 2.2.1. Inert Experimental Protocole

This study aims to measure and analyze the temperature curves of the NPs, for which a rigorous experimental protocol was developed. Furthermore, we have set a globalized initial condition for all the samples used in all experiments to validate the maximum temperature reached.

##### Sample Preparation

The NPs sample consisted of a 500 μL sample of SPIONs placed inside a 177.8 mm × 5 mm tube. The tube is placed for 90 s in the ultrasonic bath, set at 37 °C, to neutralize any aggregation that might have occurred. Afterward, the sample is placed in the calorimeter. Note that the sample must coincide at the very middle of the coil, which is why the sample holder and the coil’s initial position are fixed accordingly. Although the external temperature is isolated from the sample, the water temperature continuously goes through the sample holder, which affects the sample’s temperature, which is set at 37 °C to represent the temperature of the human body. Therefore, after placing the sample inside the holder, the temperature will be stabilized at 37 °C. This procedure is conducted in all the experiments, creating almost the same initial condition and initial temperature. Hence, the fiber optic cables are connected, and we wait until the sample temperature is stabilized at 37 °C.

##### Exposure to AMF

Once the fiber optics measure the sample temperature at 37 °C, the fiber optic measurement is turned off. Then, we initiate the generation of the desired AMF with the desired specifications. Simultaneously, we re-initiate the temperature data collection of the fiber optics to acquire the temperature rise curve, which will sample it every 1 s. Thus, the duration of the experiments in this study is 900 s.

##### Post Exposure

Once the 900 s are over, the AMF generation is seized, but the temperature measurement continues to plot the temperature decay curve.

#### 2.2.2. Data Processing

##### Waveform Slope Measurement

The Wavefom’s Slope is determined by:(2)$$Slope=H_{ac}\cdot f\cdot S_{f}$$
*f* (Hz) is the frequency set by the user, $H_{ac}$ (A· m${}^{-1}$) is the AMF intensity determined by the current dissipated in the coil, which the user also selects, leaving the $S_{f}$, which is the slope factor, which has to be deduced, except in the case of the SN waveform slope value, which is 2π. $S_{f}$ is a constant specific for each waveform that determines the fraction of the total period the waveform needs to rise and fall at any given AMF intensity or frequency.

The rise-time of the positive ramp, steady-state and fall-time of the negative ramp of each waveform was measured using an oscilloscope (see Figure 6), according to which the calculations were made to develop the factor of each waveform. The measurements were repeated at all frequencies to make sure the generation is stable at all frequencies. The slope factor values are represented in Table 1.

Hence, inserting the values into Equation (Equation 2), we can determine the Slope of each waveform at a given AMF intensity and frequency.

##### Thermal Response and Signal Analysis

The thermal measurements collected by the Luxetron 3300 are saved in an Excel file. The file contains the data of all three points of interest (sample, tube, and ambient), in addition to the time of each measurement. Once plotted, the data acquired from the sample sensor will allow us to deduce ΔT; however, this is not considered accurate without a mathematical adjustment. Hence, OriginPro was used to undergo the curve fitting. The first step was to introduce the fitting function:(3)$$T\;=\;T_{0}+\Delta T\left(1-e^{\left(\frac{-t}{\tau_{T}}\right)}\right)$$
where ΔT is the temperature variation, $\tau_{T}$ is the characteristic time of the curve, *t* is time, $T_{0}$ is the initial temperature and *T* is the measured temperature value. The fitting deduces ΔT and $\tau_{T}$, but the user can also introduce a fixed value.

In the first round of temperature curve fitting, we used OriginPro to automatically determine the adequate fitting of each temperature curve and generate its corresponding $\tau_{T}$. The objective was to compare the $\tau_{T}$ of all the experiments and find out if a single $\tau_{T}$ value can be deduced to fit all the temperature curves. However, after conducting the curve fitting of all the experiments, we compared the $\tau_{T}$ values acquired by OriginPro to determine how significant the value variation was, and unfortunately, it was pretty substantial. This presented an issue because as long as we do not have a $\tau_{T}$ valid for all the experimental curves fitting, the mathematical model was not considered valid. Hence, a single $\tau_{T}$ value had to be deduced using all the $\tau_{T}$ we acquired from the automatic fittings. Our approach is called cast-outapproach and consisted of the following steps:Conduct the first fitting where the $\tau_{T}$ is generated.Calculate the average $\tau_{T}$ value and its error (σ).Set a $\tau_{T}$ interval [$\tau_{T}$ + (*x*.σ), $\tau_{T}$ - (*x*.σ)] with the average value in the center. All the values outside this interval are cast out.Then, use a smaller *x* value for the interval, and repeat the cast-out process until no values are cast out in two consecutive *x*.σ values.When a $\tau_{T}$ is decided, repeat all the fittings with the fixed $\tau_{T}$ value, due to which ΔT is determined and used for further studies.

The graphs in Figure 7 represent the data of the temperature sensors found in the sample (in gray) and Pasteur Pipette (in red). The third sensor’s measurement was not included in the graphs since it is an ambient temperature sensor placed in a water cup measuring a fixed value at 23 °C (see Figure 4). We have four different graphs; all of them are measurements of the TP signal. However, the two on top are experiments conducted at 100 kHz and 3.2 mT, whereas the two on the bottom are conducted at 500 kHz 1.07 mT. As for the fitting curves, the green fitting is the one made using the $\tau_{T}$ generated by OriginPro, whereas the cyan fitting is the one made using the $\tau_{T}$ chosen following the cast-out approach. As we can see, the difference between the generated and fixed $\tau_{T}$ was insignificant, which allowed us to move forward and use the fixed $\tau_{T}$ value (in cyan) on the rest of the temperature curves. All the fittings showed slight differences between their generated $\tau_{T}$ fitting and the fixed $\tau_{T}$ fitting, which led us to conclude that the fixed $\tau_{T}$ is valid.

Hence, the $\tau_{T}$ chosen at the end was equal to (408.80722±15.46%) s, where all the experimental results that did not get cast out in the cast-out approach were considered valid and included in this study. Furthermore, the mathematical model enhanced our measured data reading. For example, in the previous graph, the ΔT value was adjusted from 3.43 °C to 3.28 °C thanks to the mathematical model, which was the case for all the experimental results.

On the other hand, the fitting of the entire temperature curve was to counter the severe distortion effect seen in the measurements at t=0, which made the polynomial and linear fittings somewhat inaccurate. Conveniently, the constant increase in the thermal power through time made the curve fitting the entire exposure period easier. Furthermore, since the same temperature evolution Equation (Equation 3) was used, the fitting method used in this study was homogeneous with the Lucas fitting, allowing a more accurate fitting, as seen in Figure 7, where the distortion effect was neutralized due to the fitting’s noise filtering capability.

##### Normalized Thermal Power Increment $\Delta P^{*}$

In a previous study conducted in our laboratory [27], the calorimeter was calibrated by deducing its thermal conductivity ($C_{D}$), thermal capacity ($C_{T}$) and the characteristic time of the curve ($\tau_{T}$). These values are vital to determining the NPs SAR.

However, the purpose of this article is to determine a relationship between the thermal power of signals with different slopes; hence, we will use the calculation in Rosales et al. [27] to establish this relationship.

As we previously mentioned, the thermal conductivity is always similar at a specific point [27,31]. Since the fiber optic sensor is always placed at the sample center, we can assume that the thermal conductivity is constant. Therefore, the thermal conductivity in Rosales et al. [27] is deduced from the following temperature increment formula:(4)$$\Delta T\left(\infty \right)=\frac{P_{W}}{C_{D}}$$
where ΔT(∞) is the temperature increment, $P_{W}$ is the thermal power and $C_{D}$ is the thermal conductivity. By dividing the thermal difference of 2 different experiments, we will obtain:(5)$$\frac{\Delta T{\left(\infty \right)}_{i}}{\Delta T{\left(\infty \right)}_{j}}=\frac{{P_{W}}_{i}}{{P_{W}}_{j}}$$

However, in the case of similar thermal increments, the ratio would be equal to 1, which is why a −1 is added to neutralize the $\Delta T\left(\infty \right)$ equality effect, which will now give us:(6)$${\Delta P}_{i,j}^{*}=\frac{\Delta T{\left(\infty \right)}_{i}}{\Delta T{\left(\infty \right)}_{j}}-1=\frac{P_{Wi}}{P_{Wj}}-1$$
where $\Delta P^{*}$ is the normalized thermal power increment between 2 different waveforms.

## 3. Results

The experiments were conducted in a way that allowed us to see the effect of the AMF strength, frequency and NP concentration on the heating efficiency while placed under conventional and conventional AMF waveforms. Then, the thermal power increment between the conventional and non-conventional signals is deduced by applying the acquired final temperatures in Equation (Equation 6). Finally, to validate the relationship between the power dissipation and the waveform slope, Equation (Equation 6) is adjusted to Equation (Equation 7) and then used to calculate the normalized power dissipation ($\%\Delta P_{SN,SN}^{*}$).

### 3.1. Magnetic Field Effect

The first thing that needed to be confirmed was the proportional effect of the magnetic field on the final temperature. However, the impact of the waveform is not yet determined, so this section will help us determine the effect of the signal slope on the NP heat efficiency. Hence, using 75 mgFe/mL NPs at 100 kHz frequency, the AMF strength was variated from 0.5 mT to 3.3 mT. The results are shown in Figure 8.

We noticed a dominance made by the signals with the higher slopes in heat efficiency over the rest of the waveforms. Furthermore, the higher the field intensity, the more significant the difference is. On the other hand, a conventional SN signal is always in third place behind the TS and TP waveforms, respectively, except at 2.3 mT, where the TT signal marked a slightly higher value than the SN. However, TR signals have trails in heat efficiency throughout all the experiment’s magnetic field intensities.

The results of the experiments of this section proved to us the proportionality of the NP heating efficiency with the AMF strength. Furthermore, they showed us that TS and TP signals have higher heating capability than the SN signal. However, TP unexpectedly exceeded TS at 0.5 and 1.07 mT, scoring a slightly higher value of Δt. This phenomenon is currently being investigated to determine if it occurs for specific reasons or because of the weak magnetic field intensity.

### 3.2. Frequency Effect

The experiments were conducted at four different frequencies and five AMF intensities to visualize the frequency effect on the NP’s heat efficiency. The objective is to see if we will see an increase in NP’s thermal performance by fixing the nanoparticle volume and concentration and varying the frequency. Figure 9 shows the results of the inert experiments conducted at 100, 200, 500 kHz and 1 MHz frequency, using AMF intensities of 3.21 (in blue), 2.14 (in green), 1.60 (in magenta), 1.07 (in orange) and 0.53 (in red). Then, a connecting line was drawn between the ΔT corresponding to the same AMF intensities at different frequencies, which allows the visualization of any variation in the NPs’ thermal power.

However, the coil suffered from overheating at higher frequencies, which is why Figure 9 shows many inconsistencies in the experimental results. Furthermore, the orange color in the colormap has two AMF intensities, where the 0.89 mT was used only for the TS 500 kHz frequency experiment instead of the 1.07 mT because TS was unable to reach the latter AMF intensity due to severe thermal losses.

Nevertheless, the results still showed a linear increase in all AMF intensities for all signals. For example, this is seen in curves *A* and *B*, which represent the 2.14 mT AMF intensity (in green) of the SN and TR waveforms, respectively, at 100, 200 and 500 kHz. On the other hand, 3.21 mT (blue) shows a sharper increase, but unfortunately, the experiments were only conducted at 100 and 200 kHz frequencies. In contrast, a less intense growth is witnessed at 1.60 and 1.07 mT intensities, whereas at 0.53 mT, a moderate increase is seen throughout all four applied frequencies.

The coil overheating resulted in the discontinuity of many experiments, resulting in the absence of waveforms through the applied frequencies. Which is why we decided to represent the highest AMF intensity that was successfully experimented on all waveforms at each of the four frequencies. Figure 10 represents the results of the 1 MHz experiments at 0.535 mT (equivalent to 5 $A_{pk-pk}$ output AC), the 500 kHz experiments at 1.07 mT except for the TS signal, which, due to heavy coil heating, was limited to 0.89 mT (equivalent to 10 $A_{pk-pk}$ and 8.4 $A_{pk-pk}$ output AC, respectively), the 200 kHz experiments at 2.14 mT (equivalent to 20 $A_{pk-pk}$ output AC) and the 100 kHz experiments at 3.21 mT AMF field intensity (equivalent to 30 $A_{pk-pk}$ output AC).

The 100 kHz (in black) results are similar to the maximum values in Figure 8, yet the Δ*T* value of TS at 200 kHz (in Red) /2.14 mT is almost equal to the value of TS at 100 kHz/3.21 mT. At 500 kHz (in green), although the TS was placed at 0.890 mT lower than the rest of the waveforms, it still managed to reach a higher heat efficiency than the conventional AMF signal. However, at 1 MHz (in blue), the values are almost equal, but the conventional signal’s scored the highest final heating temperature.

The results of the experiments of this section proved to us that the proportionality of the NP heating efficiency with the AMF frequency. Similar to the previous section, they showed us that TS and TP signals have higher heating capability than the SN signal at higher frequencies (200 kHz and 500 kHz). However, at frequencies as high as 1 MHz, the TS and TP lose their dominance to the SN, which is also unexpected. In addition, the increase in the frequency effect at 0.5 mT is significantly decreased compared to 3.21 mT. It is unfortunate that the coil heating prohibited us from conducting experiments at a *B* higher than 0.5 mT since we already witnessed irregularities at a magnetic field strength this low in the previous section on the one hand, and we were not able to see the thermal capability of high AMF intensities at high frequencies on the other.

### 3.3. Concentration Effect

The effect of the concentration has been studied in tow suspensions at 75 mgFe/mL and another obtained by diluting the latter on at 38.6 mgFe/mL. Therefore, in this section, we will determine the concentration effect on NP heating. Understanding the impact the concentration has on the results is vital, especially for the in vitro phase, where the toxicity factor will be taken into consideration.

Hence, we repeated the experiments of the previous section using the same frequencies, except for the 200 kHz experiments (in red) where the experiments were conducted at both 2.09 and 3.21 mT (equivalent to 20 and 30 $A_{pk-pk}$ output AC) except for TS, which was applied at only 2.09 mT, due to the sever coil overheating that was suffered at 3.21 mT and kept interrupting the experiments. The results are shown in Figure 11.

The results showed a decrease by half in the heat efficiency at 100 kHz (in black) compared to the 75 mgFe/mL experiments. At 200 kHz (in red), it required increasing *B* to 3.2 mT to reach the same values the 75 mgFe/mL concentration had achieved at 100 kHz in the previous section. Hence, by decreasing the NP concentration by half, the heat efficiency also decreased by half. Since the decrease is linear, this means that there is no aggregation occurring in the experiments using 75 mgFe/mL concentration. Nevertheless, even if aggregation occurred, it would be identical in all the experiments since the same methodology was used throughout the study; hence, it would be hard to determine whether the AMF shape effects the NP aggregation.

On the other hand, TS followed by TP both showed a higher heat production than SN, as expected. On the other hand, at 500 kHz (in green), the final values are slightly lower than those achieved using 75 mgFe/mL NPs. However, still, the TS shows higher efficiency than SN, although the former was placed 0.890 mT lower than the latter. However, at 1 MHz (in blue), we can see TP and TS regaining their superiority in heat efficiency compared to SN. The latter is even surpassed by TR and TT waveforms. Nevertheless, TP showing higher efficiency than TS is also unexpected, an irregularity seen in all the experiments conducted at *B* equal to 0.5 mT.

The results of the experiments of this section confirmed the positive effect of the frequency on NP heating. It also showed that decreasing the NP concentration negatively affects lower frequencies (100 kHz and 200 kHz), although this effect is reduced significantly at 500 kHz. The results at 1 MHz, although irregular, still showed less clustering of Δt values compared to the 75 mgFe/mL experiments. This could mean that lower concentrations are more responsive to unconventional waveforms at high frequencies, where a high concentration is more responsive to the conventional signal. Yet again, we cannot jump to conclusions until further investigation is conducted on the NPs’ behavior at very low magnetic field intensities.

### 3.4. Thermal Power Ratio

The power dissipation of the NPs is calculated using Equation (Equation 1). However, when comparing the $\tau_{T}$ deduced from the calorimeter calibration in Rosales et al. [27] to the $\tau_{T}$ we deduced using the error σ approach, we found a significant difference. By comparing the fitting conducted using our $\tau_{T}$ in Figure 7 (in clear blue) to the one conducted using the $\tau_{T}$ from the calibration in Figure 12 (in violet), it can be seen that the $\tau_{T}$ acquired from the calibration was not reliable. Hence, until we have enhanced the calorimeter calibration and acquire a $\tau_{T}$ almost equal to the $\tau_{T}$ we deduced in the previous sections, the Power dissipation calculation will not be reliable. Therefore, we decided to use a different approach to analyze the heating capabilities of each signal.

Using Equation (Equation 6), we were able to establish a ratio determining the unconventional waveform’s efficiency compared to the conventional sinusoidal, where a negative and a positive $\Delta P_{i,j}^{*}$ refer to a lesser and higher efficiency, respectively. Furthermore, the SAR value was calculated based on the Box-Lucas method described in Wildeboer et al. [31], using the $\tau_{T}$ deduced from the cast-outapproach described previously. However, the thermal capacitance $\left(C_{T}\right)$ was taken from the calibration results in Rosales et al. [27], although the value is probably inaccurate since the $\tau_{T}$ acquired from the same calibration was also unreliable. However, similar to the $\Delta P_{i,j}^{*}$, we merely aim to analyze and compare the SAR values of each signal. The results are represented in Table 2.

In both concentrations found in Table 2, the TS and TP were more efficient than the SN waveform in the entire experimental conditions except at 1 MHz while using 75 mgFe/mL NPs. Even then, SN had a thermal power more efficient by only 5.12% compared to TS and 3.4% in the case of TP. TS dominated SN in the rest of the experiments, achieving maximum efficiency at 200 kHz/2.1 mT, with $\%\Delta P_{i,j}^{*}$ equal to 74.45% at both 75 mgFe/mL and 38.6 mgFe/mL. These results prove that AMF waveforms with higher slopes have a higher impact on the NP’s heating efficiency and the overall MNP hyperthermia technique.

### 3.5. Power Dissipation Dependency on the Waveform’s Slope

In order to determine the dependency of NPs’ power dissipation on the signal slope, we have to go back to the Rosensweig equation in Equation (Equation 1). We set our focus upon the magnetic field intensity $H_{ac}$ and the frequency *f* while neglecting the other variable. The thermal power depends on the product of *f* and $H_{ac}$ to the power two, which means if normalized by dividing to the product of *f*·$H_{ac}$ to the power two, the thermal power should be essentially almost constant since the NP denominator τ varies slightly with the frequency. Next, we go back to the Slope formula in Equation (Equation 2), and we also focus on the same variables. Looking at these two equations, we notice that by dividing the thermal power from Equation (Equation 1) by the product of Slope·$S_{f}^{-1}$ from Equation (Equation 2) to the power of two, we can determine the relationship between the former and the waveform slope.

Since we do not possess an actual thermal power value, we will use the $\%\Delta P_{i,j}^{*}$ from Table 2. However, zero and negative values cannot be included in the calculations, so instead of subtracting −1 in Equation (Equation 6), we carried it out as follows:(7)$${\%\Delta P}_{i,SN}^{*}=\frac{\Delta T{\left(\infty \right)}_{i}}{\Delta T{\left(\infty \right)}_{SN}}+100$$

The objective in Equation (Equation 7) is to set $\%\Delta P_{SN,SN}^{*}$ at 100 instead of 0. Next, we divide the previous result by the slope as follows:(8)$$n\%\Delta P_{i,j}^{*}=\frac{\%\Delta P_{i,SN}^{*}}{Slope^{2}\cdot S_{f}^{-2}}$$

All the values of ΔT used in this study and represented in Figure 8, Figure 10, Figure 11 and Table 2 were applied in Equation (Equation 7) to then deduce their corresponding $n\%\Delta P_{i,j}^{*}$ from Equation (Equation 8).

The values were so small that they were plotted using a logarithmic scale plot, as shown in Figure 13.

In Figure 13, the experiment results using 75 mgFe/mL are represented in black, whereas the ones conducted using 38.6 mgFe/mL are represented in red, and every waveform used in the experiments is represented by its own sign.

Although $\%\Delta P_{i,j}^{*}$ should be almost constant or very slightly dependent on frequency, following the Rosensweig equation, it actually strongly depends on the Slope ($H_{ac}$·*f*), thatison$H_{ac}$ and *f*, as can be seen in Figure 13. Of course, this does not contradict that $\%\Delta P_{i,j}^{*}$ positively depends on the Slope and that, specifically, the Slope that keeps the frequency and magnetic field peak equal also determines the $\%\Delta P_{i,j}^{*}$ value as a function of the waveform, as shown in Figure 13. The explanation for this phenomenon still deserves further investigation and may be related to recent studies [34,35].

## 4. Discussion

The experimental results in this study showed the heat generation superiority of TS and TP compared to the SN, TT and TR signals. Moreover, in some instances, TS reached a 75% higher efficiency in thermal power generation than the SN, with TP closely behind. In Allia et al. [34] and Barrera et al. [35], theoretical studies were conducted to analyze the effect of the hysteresis loop on NPs’ heating efficiency. The simulation results showed the superiority of a square signal over the conventional sinusoidal signal. However, since generating a pure square AC signal at these frequencies is intrinsically very difficult and not possible with the equipment we have now, a more realistic simulation should be conducted in the future to compare the results with the experimental ones. Furthermore, an intensive theoretical study and simulations will focus on the consequences of replacing a single SN term with the Fourier sum of the unconventional signals, in addition to the analysis of the waveforms’ Fourier spectrum and determining the latter’s relationship with the frequency and relaxation time dependence on the dissipated power from Equation (Equation 1).

The results showed the effect of the frequency on small NPs, as mentioned in Roca et al. [16], where a temperature increase was witnessed even at small magnetic field strengths. Furthermore, the experiments showed that a 38.6 mgFe/mL NP concentration is more responsive to unconventional than conventional AMF waveforms at 1 MHz frequencies, where all waveforms exceeded SN in heat dissipation by 55.46% at best by TP and by 9.67% at worst by TR. On the other hand, at lower frequencies, the concentration of the MNPs were proportional to their heating efficiency. This fact led to the assumption that no aggregation occurred during the experiments since the lowering of the NP concentration by half led to the loss of almost half of the heat efficiency, which showed the adequacy of the experimental protocols used in these experiments. In contrast, abnormal phenomena appeared at magnetic field strengths below 1.5 mT, where NP showed unexpected ΔT results. This might be due to the chains’ formation effect discussed in Mille et al. [21]. This should be investigated further to determine if the cause is the exposure time, *B*, to what extent the MNPs’ concentration has an effect on the formation of the chains, and if the waveform shape has any impact on it. Furthermore, it is vital to determine whether the AMF shape affects the NP aggregation and to identify each waveform’s affect magnitude for different NP concentrations, frequencies and AMF strength conditions.

These results confirm the conclusion in Rosales et al. [27] that using a low SPION concentration and exposing them to AMF waveforms with more significant slopes, such as TS and TP, at high frequencies has a higher efficiency than those with higher frequencies conventional signals. This will result in MNP hyperthermia therapies with lower NP toxicity risk and lower magnetic field intensity requirements. However, since the concentration effect is now determined, the shape effect should be targeted in future experiments. In Avugadda et al. [36] and Abenojar et al. [37], the cubic-shaped particles showed a higher SAR than the spherical-shaped, in addition to the promising results shown in Sánchez et al. [18], so perhaps the cubic-shaped would perform even better with non-conventional signals.

Furthermore, following the calculation of $n\%\Delta P_{i,j}^{*}$, the results showed the dependency of the thermal power generated by the NP on the waveform’s slope, as it was previously hypothesized in Garcia et al. [28], Zeinoun et al. [30] and Urbano et al. [29]. However, further investigation is required in order to determine the factor that TR and TT are aligned exponentially, while TS, TP and SN are aligned horizontally.

To summarize, the experiments showed higher heat production efficiency of TS and TP AMF waveforms at best by 71.49% and 45.73%, respectively, over the conventional SN waveform. Furthermore, the results showed that the higher the frequency, the lower concentration is required to create the most efficient conditions for the MNPs. Finally, the $n\%\Delta P_{i,j}^{*}$ calculation results proved the dependency of the power dissipation on the waveform’s slope and not only *B* and *f*, as mentioned in Rosensweig et al. [26].

## 5. Conclusions

We have successfully shown the higher efficiency of AMF waveforms of significant slopes over the conventional SN in MNP hyperthermia. The experiments showed higher heat production efficiency of TS and TP AMF waveforms at best by 71.49% and 45.73%, respectively, over the conventional SN waveform. However, it might not be the highest efficiency available, which is why further investigation will be conducted to search for a waveform that might even surpass the 71.49% superiority over SN.

Furthermore, we have demonstrated the effectiveness of the slope, frequency and NP concentration on the therapy. We concluded that the ideal conditions for low concentrations of SPIONs are the use of signals with more significant slopes (TS and TP) at radiofrequencies, which will result in increasing the efficiency of the therapy while lowering the toxicity levels.

### Future Work

The hyperthermia system requires a coil suited for radiofrequency operations in addition to an adequate cooling system for it. This will allow us to exploit the full capability of the AMF generator and conduct high-frequency experiments at higher intensities without overheating issues.

Furthermore, a glass-based in vitro sample holder is currently being designed to conduct in vitro studies since the current calorimeter is only for MR tubes. Our objective is to design a sample holder that can fit a P35Petri dish (35 mm Ø, 10 cm${}^{2}$) inside and have the ability to isolate the sample and block the external temperature interferences.

## Figures and Tables

**Figure 1 nanomaterials-11-03240-f001:**
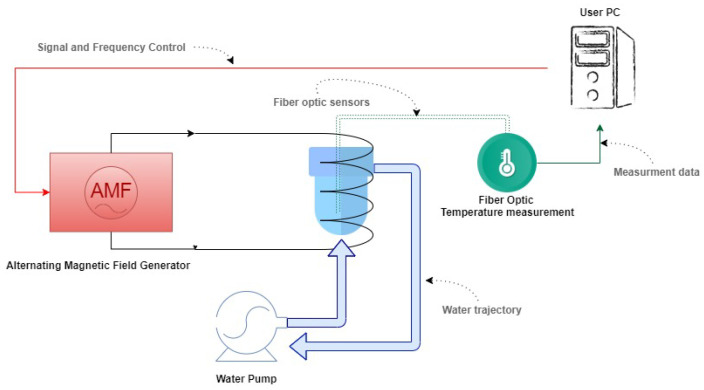
Hyperthermia system workflow.

**Figure 2 nanomaterials-11-03240-f002:**
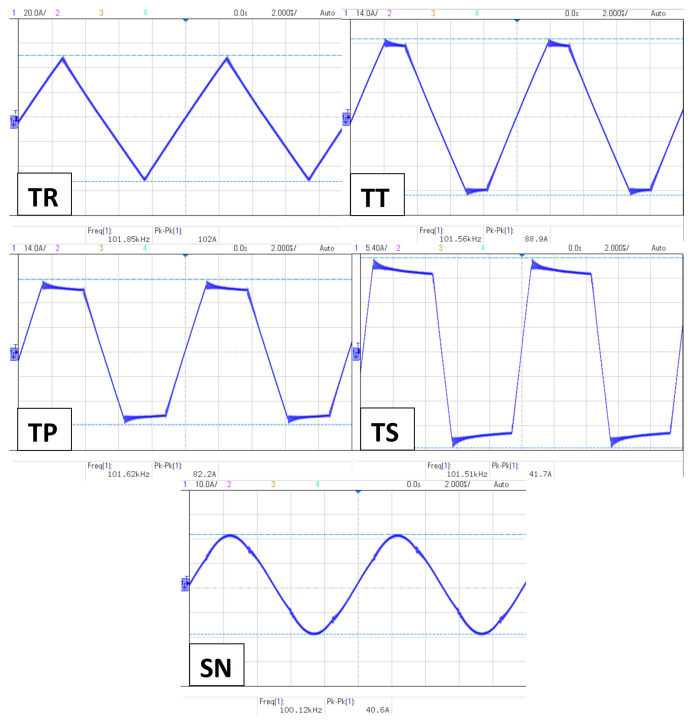
AMF generator waveforms.

**Figure 3 nanomaterials-11-03240-f003:**
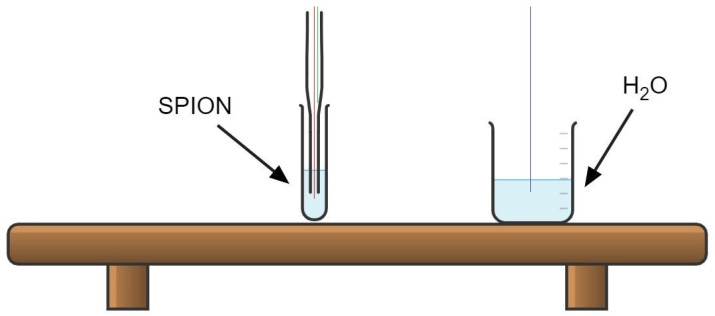
Fiber optic temperatue sensor placement.

**Figure 4 nanomaterials-11-03240-f004:**
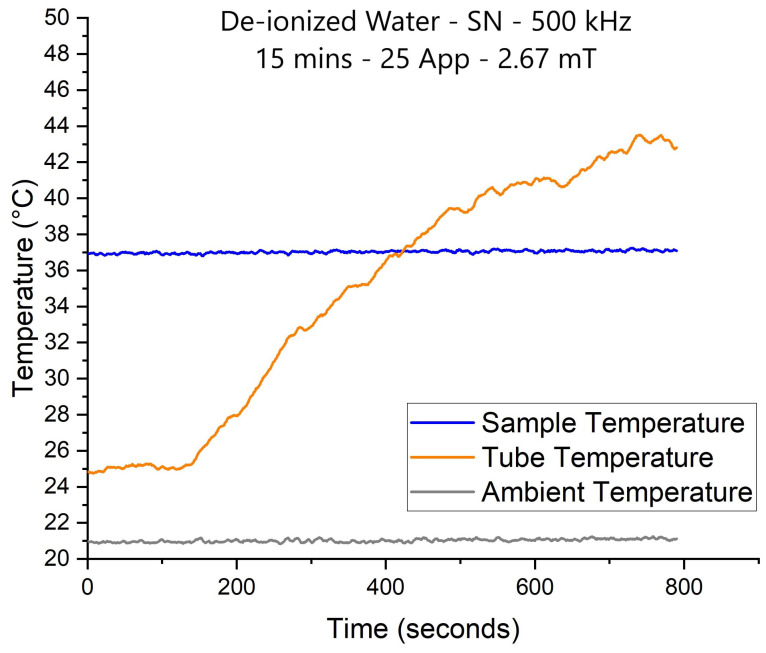
Sample temperature isolation test using de-ionized water.

**Figure 5 nanomaterials-11-03240-f005:**
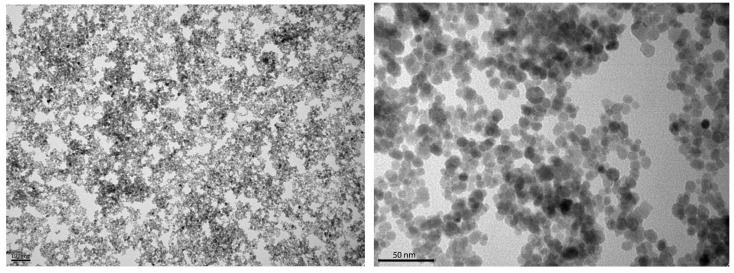
JEOL JEM 1011 TEM images of the superparamagnetic iron oxide NPs with Gatan ES1000Ww camera.

**Figure 6 nanomaterials-11-03240-f006:**
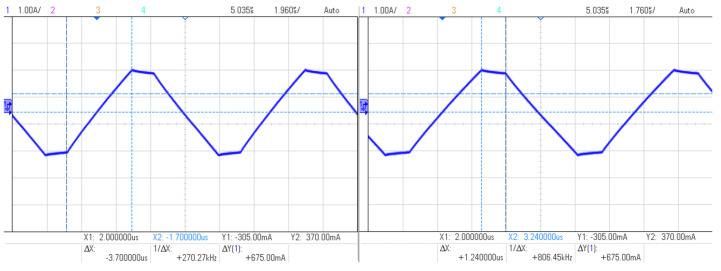
Slope measurement via oscilloscope.

**Figure 7 nanomaterials-11-03240-f007:**
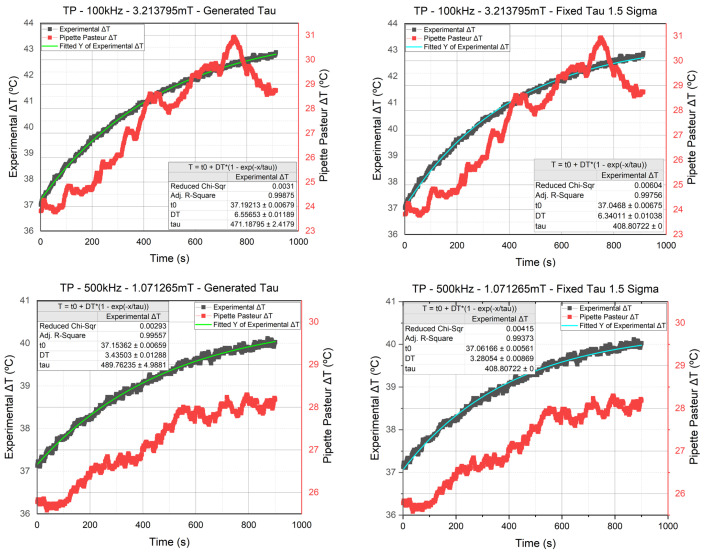
Comparing the curve fittings of the $\tau_{T}$ value generated by OriginPro (in green) to the one deduced from the cast-outapproach (in cyan) using the experimental results of the TP signal at 100 and 500 kHz.

**Figure 8 nanomaterials-11-03240-f008:**
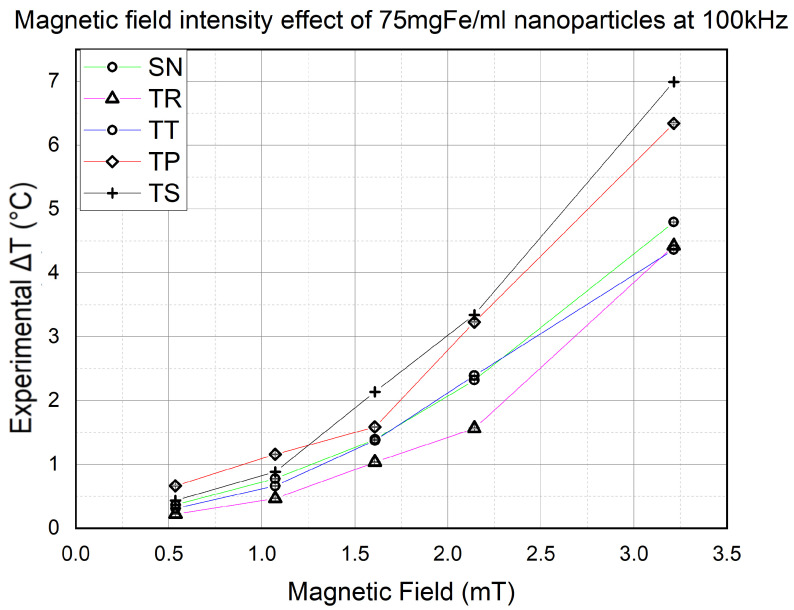
Magnetic intensity effect experiments results of 75 mgFe/mL MNPs at 100 kHz for 900 s at *B* going from 0.5 to 3.2 mT. The Error bars are barely seen due to their small value (<10${}^{-3}$).

**Figure 9 nanomaterials-11-03240-f009:**
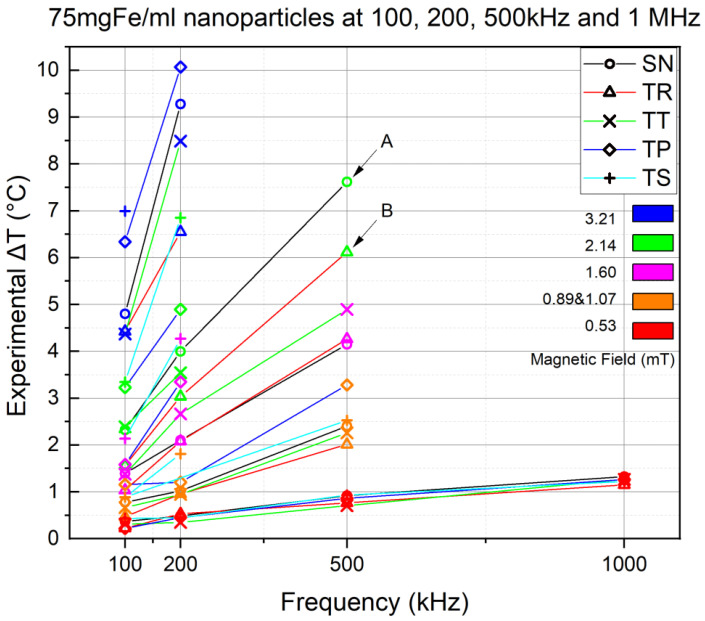
Frequency effect experiments result of 75 mgFe/mL MNPs at frequencies going from 100 kHz to 1 MHz for 900 s at *B* going from 0.5 to 3.2 mT.

**Figure 10 nanomaterials-11-03240-f010:**
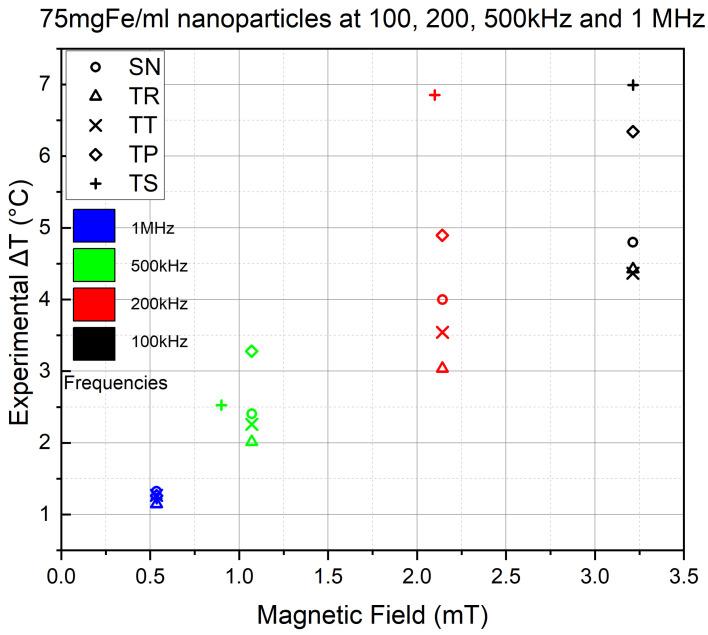
Comparing the 900 s experiment results of all signals at 0.53, 1.07 (except for TS at 0.890), 2.14 and 3.21 mT and 1 MHz, 500, 200 and 100 kHz frequency, respectively.

**Figure 11 nanomaterials-11-03240-f011:**
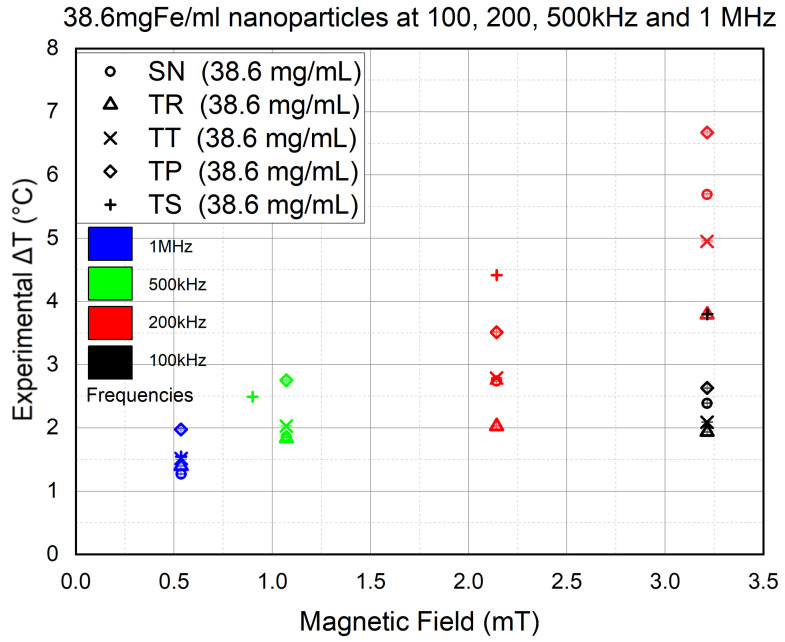
Concentration effect experiments result of 38.6 mgFe/mL MNPs at frequencies going from 100 kHz to red 1 Hz for 900 s at *B* going from 0.5 to 3.2 mT. The Error bars are barely seen due to their small value (<10${}^{-3}$).

**Figure 12 nanomaterials-11-03240-f012:**
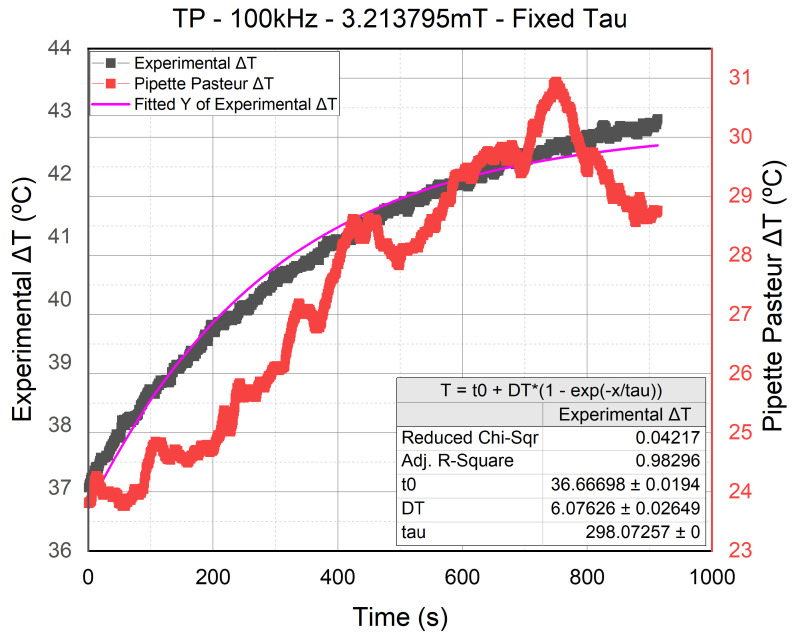
Repeating the fitting of Figure 7 while fixing $\tau_{T}$ equal to the calibration result value.

**Figure 13 nanomaterials-11-03240-f013:**
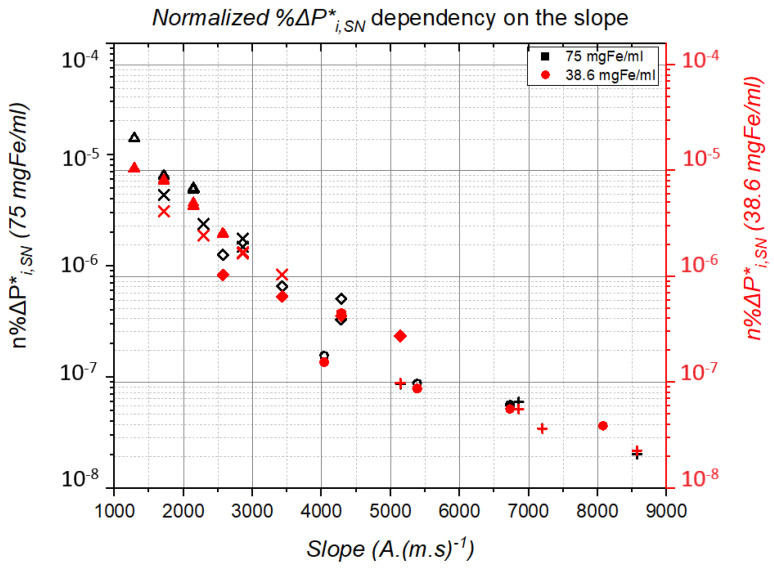
Normalized $\%\Delta P_{i,j}^{*}$ calculation plot using a logarithmic scale plot.

**Table 1 nanomaterials-11-03240-t001:** Slope factor.

Waveform	$S_{f}$
SN	1/2π
TR	1/2
TT	3/8
TP	1/4
TS	1/8

**Table 2 nanomaterials-11-03240-t002:** Thermal power increment ratio calculation of the values from Figure 8, Figure 10 and Figure 11.

75 mgFe/mL					38.6 mgFe/mL				
Waveform	AMF(mT)	ΔT (°C)	SAR (W/mg)	$\%\Delta P_{i,\mathrm{SN}}^{*}$	Waveform	AMF (mT)	ΔT (°C)	SAR (W/mg)	$\%\Delta P_{i,\mathrm{SN}}^{*}$
**100 kHz**									
**SN**	3.21	4.79	0.56	-	**SN**	3.21	2.38	0.54	-
**TR**	3.21	4.42	0.52	−7.79%	**TR**	3.21	1.93	0.44	−18.95%
**TT**	3.21	4.36	0.51	−8.98%	**TT**	3.21	2.09	0.47	−12.46%
**TP**	3.21	6.34	0.74	+32.12%	**TP**	3.21	2.63	0.60	+10.20%
**TS**	3.21	6.99	0.82	+45.73%	**TS**	3.21	3.79	0.86	+59.10%
**200 kHz**									
**SN**	2.14	3.99	0.47	-	**SN**	2.14	2.74	0.62	-
**TR**	2.14	3.03	0.35	−24.04%	**TR**	2.14	2.02	0.46	−26.15%
**TT**	2.14	3.54	0.41	−11.32%	**TT**	2.14	2.78	0.63	+1.65%
**TP**	2.14	4.89	0.57	+22.46%	**TP**	2.14	3.51	0.80	+28.09%
**TS**	2.14	6.85	0.80	+71.49%	**TS**	2.14	4.41	1.00	+61.00%
**500 kHz**									
**SN**	1.07	2.4	0.28	-	**SN**	1.07	1.85	0.42	-
**TR**	1.07	2.01	0.23	−16.24%	**TR**	1.07	1.83	0.41	−1.25%
**TT**	1.07	2.26	0.26	−5.95%	**TT**	1.07	2.02	0.46	+9.22%
**TP**	1.07	3.28	0.38	+36.5%	**TP**	1.07	2.75	0.62	+48.26%
**TS**	0.89	2.52	0.29	+5.08%	**TS**	0.89	2.49	0.57	+34.18%
**1 MHz**									
**SN**	0.53	1.32	0.15	-	**SN**	0.53	1.27	0.29	-
**TR**	0.53	1.14	0.13	−13.29%	**TR**	0.53	1.39	0.32	+9.67%
**TT**	0.53	1.26	0.15	−4.4%	**TT**	0.53	1.51	0.34	+19.44%
**TP**	0.53	1.15	0.13	−4.64%	**TP**	0.53	1.97	0.44	+55.46%
**TS**	0.53	1.22	0.14	−7.86%	**TS**	0.53	1.54	0.35	+21.52%

## Data Availability

The data presented in this study are available on request from the corresponding author.

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
