# Peer review of "Enhancing Magnetic Hyperthermia Nanoparticle Heating Efficiency with Non-Sinusoidal Alternating Magnetic Field Waveforms"

_nanomaterials, 2021, doi:10.3390/nano11123240_

Round 1

Reviewer 1 Report

Article: Enhancing Magnetic Hyperthermia Nanoparticle Heating Efficiency with Non-sinusoidal Alternating Magnetic Field Waveforms

Authors:

Michael ZEINOUN, Javier DOMINGO-DIEZ, Miguel RODRIGUEZ-GARCIA, Oscar GARCIA, Miroslav VASIC, Milagros RAMOS, Jose Javier SERRANO OLMEDO

The study presented the effect of various waveforms on the heat production effectiveness of magnetic nanoparticles, seeking to prove that signals with more significant slope values such as the trapezoidal and almost-square signals allow the nanoparticles to reach higher efficiency in heat generation. They showed with experimental results that a remarkably higher heat production performance of the nanoparticles when exposed to trapezoidal and almost-square signals than conventional sinusoidal signals. Also the nanoparticles responded better to the trapezoidal and almost square signals. They used experimental results to calculate the normalized power dissipation value and proved its dependency on the slope. This paper it is not acceptable for publication in Nanomaterials in its present form. However, with major revision it might be suitable for publication.

My comments are as follows:

Literature names throughout the article:

  • Line 23: disease. [2].
  • Line 31: to heat [3] [4].
  • Line 68 hyperthermia. [32] showed
  • Line 103, 163, …

Spelling with lower or upper case letters

  • Line 59, 291, 480, …

Spaces between text and references, units

  • Line 43: fluids[11]→fluids [11]
  • Line 57: 10nm→10 nm
  • Line 74: 72%→72 %
  • Line 80: 78kHz→78 kHz, …

Standardization of abbreviations throughout the article

  • MHT: line 66, 68, 76, 83, 93, 145,…
  • MNPs: line 38, 408, 447, 460, 474, 480
  • AMF: line 122, 145, 151
  • SAR: line 98, 287
  • NP: line 189

spelling mistakes

  • line 68: hydoxy→hydroxy
  • line 79: dependant→dependent
  • line 89: wen→when
  • under Fig. 13: logartimic→logarithmic
  • line 452: nanoparticle→nanoparticles

Abbreviations without explanation

  • Line 216: NPS
  • Line 221: SPIONs

Equalization of terms: Origin Pro↔OriginePro

Separator

  • Line 177, 436

Order of references

  • Line 137, 193, 468, …

Which water was used?

  • Line 194, Fig. 4→pure water sample (or MiliQ water) or De-ionized water?!?

Units

  • 200 seconds: line 236, 237, 239

Size of the nanoparticles?

  • What nanoparticles were used, did you dry them? Please, complete the preparation of the nanoparticles you used in the experimental work.

Which sample? (line 221)

the image caption must be consistent

  • line 196: Fig. 4
  • line 248: Fig. (2)
  • Line 273: Fig.4

correct entry Rosensweig (line 112) or Rosenweig (line 411)

I cannot find a black square (n) in Figure 12 and 13 that indicates 75 mgFe / mL.

Reviewer 2 Report

Michael ZEINOUN et al. Enhancing Magnetic Hyperthermia Nanoparticle Heating Efficiency with Non-sinusoidal Alternating Magnetic Field Waveforms

1) The affiliation for Oscar GARCIA  and Miroslav VASIC is missing;

2) Lines 117 – 119: ΔT should be here be defined as the temperature rise after long enough time;

3) Lines 165 and 195: Explain „30Apk-pk/3.2mT“;

4) Line 206 and lines 97 - 100: Since „the nanoparticles used are superparamagnetic iron oxide nano-spheres without any biofunctionalization“ while „MHT experiments demonstrate that MNPs have a decreased energy absorption per unit mass, or specific absorption rate (SAR), once internalized in cells, ranging from 90 % to 50 % depending on particle size, shape, and composition [15], [16]“ prompts the question about using no biofunctionalization;

5) Line 128 and line 245: In order to enhance the readability, it is desirable to here define the slope factor of the AMF signal;

6) In Eq. (3), t0 to be replaced with T0 and x with t (time), but then what is x defined as „a measured temperature value“?;

7) Lines 258 – 260: „After conducting a few fittings“ instead of „After conducting a few fitting values“. Actually, the whole part including lines 258 - 283 is rather confusing and must be rewritten. The differences between green and cyan curves are insignificant. In line 279, „The τT chosen at the end was equal to 408.80722 ± 15.46%“ should be replaced with „The τT chosen at the end was equal to (408.8 ± 15.5) sec“. It is not clear whether this value is used afterwards for all field shapes, i.e. slopes, and frequencies. If so, why?;

8) Line 291 - 292: ΔT(∞) is called the thermal conductivity and the temperature increment in the same sentence;

9) In Fig. 9, the magnetic field is on the abscissa, not frequencies, i.e. the field was not constant while changing the frequencies and therefore the observed temperature effect are not only due to the frequency dependence;

10) In Table 2, is it really necessary to give the values for AMF to six decimal points and to list both ΔP and %ΔP? ;

11) Following Rosensweig, the only way how the AMF shape can enter the power dissipation formula is through the relaxation equation (Eds. (7) - (9)). The authors do not discuss the consequences of replacing a single sin term with the Fourier sum representing different AMF shapes or tried to relate their experimetnal results with the theoretical predictions of Barrera et al. (ref. 24 and 25);

12) As seen in Table 2, ΔT changes slightly relative to the SN value. It is not easy to imagine that this would have any serious consequences if applied in vivo having in mind all the complications mentioned in Introduction. In addition, the relevance of the paper is diminished by not using functionalized nanoparticles. 

Overall recommendation is to do the revisions stated above and then to re-submit.

Reviewer 3 Report

- Hyperthermia using non-sinusoidal waveforms is not a new topic

  • Formula (3) is wrong, should be T=t0+ΔT(1-e^(-t/τ)) and the fitting should be for ΔT and τ, not t.
  • If you are not able to keep B constant, you should normalize the results related to the applied B (Figure 9 and Figure 10), so it is not usable.

Reviewer 4 Report

Dear authors,

I recommend the publication of the paper after major revision is made in either clarification or changes in the data analysis section, and minor revisions in language and graphs editing. Please consider the following:

1. Some typos:
    1.1 line 416: "the latter" is rather "the former" if I understand the sentence correctly, and
    1.2 line 435: "a theoretical was conducted" seems to need a subject.

2. Regarding the chain formation in AC magnetic fields (Sec.1 Introduction lines 76-85) I draw your attention on the paper: Socoliuc, V; Turcu, R: “Large scale aggregation in magnetic colloids induced by high frequency magnetic fields”, JOURNAL OF MAGNETISM AND MAGNETIC MATERIALS 500 (2020) 166348; DOI 10.1016/j.jmmm.2019.166348 where evidence is shown of ~5 microns thick and tens of microns long clusters induced by 4-12mT and 100kHz magnetic fields, from light scattering experiments. The mass weight of the clusters in the colloid grows up to 80%.

3. Triangular and trapezoidal waveforms owe their advantage over sinusoidal waveforms due to their high frequency Fourier components. The Slope parameter in your analysis actually stands for the Fourier spectrum width of your waveforms (TR, TT, TP and TS).
    3.1 In Fig.2 please show the Fourier spectrum of each waveform.
    3.2 In Fig.2 please indicate the waveform acronyms.
    3.3 Discuss the Fourier spectrums in relation with the frequency and relaxation time dependence of dissipated power from Eq.1.

4. Figs.12 and 13:
    4.1 The ordinate axes' labels should contain the measured quantity and unit and not the experimental parameter (concentration),
    4.2 add the waveforms legend to Fig.13,
    4.3 you may consider dropping Fig.12 since Fig.13 does a better job.

5. Sec.3.5 Power dissipation dependency on the waveform’s slope:
    5.1 lines 413-416: It is not clear why Slope*Sf would be the divisor parameter. If one eliminate Hac from eqs.1 and 2, the parameter should be (Slope/Sf)^2. Please clarify this issue.
    5.2 Fig.12 and 13 show a decrease of heating efficiency with increasing slope. This contradicts eq.1, since increasing slope implies increasing high frequency harmonics amplitude which in turn increases the power.
    5.3 Sec.3.5 is overall unclear.

Best regards,

Round 2

Reviewer 1 Report

I believe the manuscript has been
sufficiently improved to warrant publication in Nanomaterials.

Reviewer 2 Report

The affiliation for Oscar GARCIA and Miroslav VASIC still missing.

It is OriginPro, not OriginePro.

Reviewer 4 Report

Dear authors,

I regret to inform you that I decided to not recommand you paper for publication for the following reasons:

1. Sec.3.5: As I previously mentioned, the division parameter in eq.8 is (Slope / Sf)^2. You didn't even bother to do the elementary math.

2. Eq.7: Eq.6 represents relative change. That's ok, but changing the "-1" from eq.6 to "+1" (+100 in percentages) in eq.7 simply makes no sense, 
other than, maybe, concealing the uncomfortable negative DeltaP's from Table 2.

3. Fig.13: Logarithmic data representation doesn't mean you represent the logarithm of the data.

4. Fig.13: There's a 3 orders of magnitude diminishing in n%DeltaP over 1 order of magnitude increase in Slope. This means that %DeltaP is close to 
independent of Slope, therefore your conclusion, based on, like you said "raw results", is not supported. Fourier analysis will help.

5. Line 320; "difference" -> "ratio"

Best regards,
